# ENHANCING CROSS-LINGUAL TRANSFER BY MANIFOLD MIXUP

**Huiyun Yang**[1]**, Huadong Chen**[1]**, Hao Zhou**[*][1]**, Lei Li**[2]
[1]ByteDance AI Lab
[2]University of California, Santa Barbara
{yanghuiyun.11,chenhuadong.howard, zhouhao.nlp}@bytedance.com
leili@cs.ucsb.edu

## ABSTRACT

Based on large-scale pre-trained multilingual representations, recent cross-lingual transfer methods have achieved impressive transfer performances. However, the performance of target languages still lags far behind the source language. In this paper, our analyses indicate such a performance gap is strongly associated with the cross-lingual representation discrepancy. To achieve better cross-lingual transfer performance, we propose the cross-lingual manifold mixup (X-MIXUP) method, which adaptively calibrates the representation discrepancy and gives compromised representations for target languages. Experiments on the XTREME benchmark show X-MIXUP achieves 1.8% performance gains on multiple text understanding tasks, compared with strong baselines, and reduces the cross-lingual representation discrepancy significantly.

## 1 INTRODUCTION

Many natural language processing tasks have shown exciting progress utilizing deep neural models. However, these deep models often heavily rely on sufficient annotation data, which is not the case in the multilingual setting. The fact is that most of the annotation data are collected for popular languages like English and Spanish (Ponti et al., 2019; Joshi et al., 2020), while many long-tail languages could hardly obtain enough annotations for supervised training. As a result, cross-lingual transfer learning (Prettenhofer & Stein, 2011; Wan et al., 2011; Ruder et al., 2019) is crucial, transferring knowledge from the annotation-rich *source language* to low-resource or zero-resource *target languages*. In this paper, we focus on the zero-resource setting, where labeled data are only available in the source language.

Recently, multilingual pre-trained models (Conneau & Lample, 2019; Conneau et al., 2020a; Xue et al., 2020) offer an effective way for cross-lingual transfer, which yield a universal embedding space across various languages. Such universal representations make it possible to transfer knowledge from the source language to target languages through the embedding space, significantly improving the transfer learning performance (Chen et al., 2019; Zhou et al., 2019; Keung et al., 2019; Fang et al., 2020). Besides, Conneau et al. (2018) proposes *translate-train*, a simple yet effective cross-lingual data augmentation method, which constructs pseudo-training data for each target language via machine translation. Although these works have achieved impressive improvements in cross-lingual transfer (Hu et al., 2020; Ruder et al., 2021), significant performance gaps between the source language and target languages still remain (see Table 1). Hu et al. (2020) refers to the gap as the *cross-lingual transfer gap*, a difference between the performance on the source and target languages.

To investigate how the cross-lingual transfer gap emerges, we perform relevant analyses, demonstrating that transfer performance correlates well with the *cross-lingual representation discrepancy* (see Section 3 for details). Here the cross-lingual representation discrepancy means the degree of difference between the source and target language representations in the universal embedding space. As shown in Figure 1(a), in translate-train, the representation distribution of Spanish almost overlaps

---

[*]Corresponding author.
Code is available at https://github.com/yhy1117/X-Mixup.

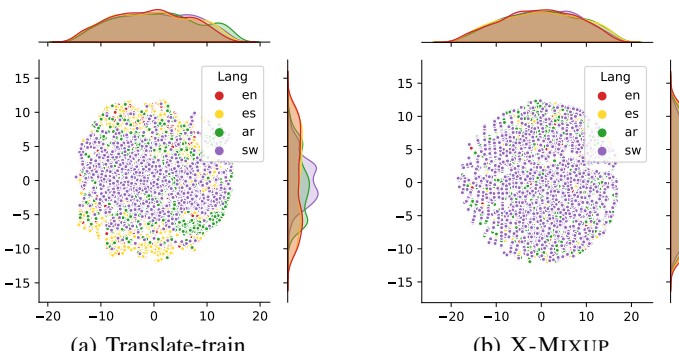

(a) Translate-train     (b) X-Mixup

Figure 1: Representation visualization of four languages: English (en), Spanish (es), Arabic (ar) and Swahili (sw) based on XLM-R. We plot the sentence representation of the XNLI test set, which is parallel across 15 languages. We average hidden states of the last layer to get sentence representations and implement the dimensionality reduction by PCA. Obviously, the cross-lingual representation discrepancies are large in translate-train, but X-Mixup reduces the discrepancy significantly.

with English, while Arabic shows a certain representation discrepancy compared with English and Swahili performs larger discrepancy, where translate-train achieves 88.6 accuracy on English, 85.7 on Spanish, 82.2 on Arabic and 77.0 on Swahili. Intuitively, a larger representation discrepancy could lead to a worse cross-lingual transfer performance.

In this paper, we propose the *Cross-Lingual Manifold Mixup* (X-Mixup) approach to fill the cross-lingual transfer gap. Based on our analyses, reducing the cross-lingual representation discrepancy is a promising way to narrow the transfer gap. Given the cross-lingual representation discrepancy is hard to remove, X-Mixup directly faces the issue and explicitly accommodates the representation discrepancy in the neural networks, by mixing the representation of the source and target languages during training and inference. With X-Mixup, the model itself can learn how to escape the discrepancy, which adaptively calibrates the representation discrepancy and gives compromised representations for target languages to achieve better cross-lingual transfer performance. X-Mixup is motivated by robust deep learning (Vincent et al., 2008), while X-Mixup adopts the mixup (Zhang et al., 2018) idea to handle the cross-lingual discrepancy.

Specifically, X-Mixup is designed upon the translate-train approach, faced with the exposure bias (Ranzato et al., 2016) problem and data noise problem. During training, the source sequence is a real sentence and the target sequence is a translated one, while situations are opposite during inference. Besides, the translated text often introduces some noises due to imperfect machine translation systems. To address them, we further impose the *Scheduled Sampling* (Bengio et al., 2015) and *Mixup Ratio* in X-Mixup to handle the distribution shift problem and data noise problem, respectively.

We verify X-Mixup on the cross-lingual understanding benchmark XTREME (Hu et al., 2020), which includes several understanding tasks and covers 40 languages from diverse language families. Experimental results show X-Mixup achieves 1.8% performance gains across different tasks and languages, comparing with strong baselines. It also reduces the cross-lingual representation discrepancy significantly, as Figure 1(b) shows.

## 2 Related Work

**Multilingual Representation Learning**  Recent studies have demonstrated the superiority of large-scale pre-trained multilingual representations on downstream tasks. Multilingual BERT (mBERT; Devlin et al., 2019) is the first work to extend the monolingual pre-training to the multilingual setting. Then, several extensions achieve better cross-lingual performances by introducing more monolingual or parallel data and new pre-training tasks, such as Unicoder (Huang et al., 2019), XLM-R (Conneau et al., 2020a), ALM (Yang et al., 2020), MMTE (Siddhant et al., 2020), InfoXLM (Chi et al., 2020), HICTL (Wei et al., 2020), ERNIE-M (Ouyang et al., 2020), mT5 (Xue et al., 2020), nmT5 (Kale

Table 1: Cross-lingual transfer performances of POS and NER tasks on languages with different data resources or different language families, where there are only labeled training data in English. The data resource refers to the resource of each language utilized in the pre-training process. For the language family, English belongs to the Germanic languages, so we divide languages into two types: Germanic one and others. Results show high-resource languages outperform low-resource ones significantly and languages dissimilar to the source language tend to perform worse.

| Language Type | | Source | Language Resource | | | | | | Language Family | | | | | |
| --- | --- | --- | --- | --- | --- | --- | --- | --- | --- | --- | --- | --- | --- | --- |
| | | | High-resource | | | Low-resource | | | Germanic | | | Other | | |
| Language | | en | es | it | pt | eu | kk | mr | af | de | nl | ar | hi | ja |
| mBERT | POS | 95.5 | 86.9 | 88.4 | 86.2 | 60.7 | 70.5 | 69.4 | 86.6 | 85.2 | 88.6 | 56.2 | 67.2 | 49.2 |
| | NER | 85.2 | 77.4 | 81.5 | 80.8 | 66.3 | 45.8 | 58.2 | 77.4 | 78.0 | 81.8 | 41.1 | 65.0 | 29.0 |
| XLM-R | POS | 96.1 | 88.3 | 89.4 | 87.6 | 72.5 | 78.1 | 80.8 | 89.8 | 88.5 | 89.5 | 67.5 | 76.4 | 15.9 |
| | NER | 84.7 | 79.6 | 81.3 | 81.9 | 60.9 | 56.2 | 68.1 | 78.9 | 78.8 | 84.0 | 53.0 | 73.0 | 23.2 |

et al., 2021), AMBER (Hu et al., 2021) and VECO (Luo et al., 2021). They have been the standard backbones of current cross-lingual transfer methods.

**Cross-lingual Transfer Learning**   Cross-lingual transfer learning (Prettenhofer & Stein, 2011; Wan et al., 2011; Ruder et al., 2019) aims to transfer knowledge learned from source languages to target languages. According to the type of transfer learning (Pan & Yang, 2010), previous cross-lingual transfer methods can be divided into three categories: instance transfer, parameter transfer, and feature transfer. The cross-lingual transferability improves a lot when engaged with the instance transfer by translation (i.e. translate-train, translate-test) or other cross-lingual data augmentation methods (Singh et al., 2019; Bornea et al., 2020; Qin et al., 2020; Zheng et al., 2021). Chen et al. (2019) and Zhou et al. (2019) focus on the parameter transfer to learn a share-private model architecture. Besides, other works implement the feature transfer to learn the language-invariant features by adversarial networks (Keung et al., 2019; Chen et al., 2019) or re-alignment (Libovický et al., 2020; Zhao et al., 2020). X-MIXUP utilizes both the instance transfer and feature transfer, which is based on the translate-train data augmentation approach and implements the feature transfer by cross-lingual manifold mixup.

**Mixup and Its Variants**   Mixup (Zhang et al., 2018) proposes to train models on the linear interpolation at both the input level and label level, which is effective to improve the model robustness and generalization. Generally, the interpolated pair is selected randomly. Manifold mixup (Verma et al., 2019) performs the interpolation in the latent space by conducting the linear combinations of hidden states. Previous mixup methods (Chen et al., 2020; Jindal et al., 2020) focus on the monolingual setting. However, X-Mixup focuses on the cross-lingual setting and faces many new challenges (see Section 4 for details). Besides, in contrast to previous mixup methods, X-MIXUP mixes the parallel pairs, which share the same semantics across different languages. As a result, the choice of parallel pairs for interpolation can build a smart connection between the source and target languages.

## 3   ANALYSES OF THE CROSS-LINGUAL TRANSFER PERFORMANCE

In this section[1], we concentrate on the cross-lingual transfer performance and find it is strongly associated with the cross-lingual representation discrepancy. Firstly, we observe the cross-lingual transfer performance on different target languages and propose an assumption. Then we conduct qualitative and quantitative analyses to verify it.

Although previous studies (Hu et al., 2020; Ruder et al., 2021) have shown impressive improvements on cross-lingual transfer, the cross-lingual transfer gap is still pretty large, more than 16 points in Hu et al. (2020). Furthermore, results in Table 1 show **the performance of low-resource languages and dissimilar languages fall far behind other languages in cross-lingual transfer tasks.**

Compared with English, the representations of other languages, especially low-resource languages, are not well-trained (Lauscher et al., 2020; Wu & Dredze, 2020), because high-resource languages dominate the representation learning process, which results in the cross-lingual representation

---

[1]In our analyses, we take English as the source language, and the dissimilar language is the language which is dissimilar to English.

Table 2: Spearman's rank correlation $\rho$ between the CKA score and cross-lingual transfer performance on two XTREME tasks, where $\dagger$ denotes training on the source language, and $\ddagger$ denotes the translate-train approach. $*$ denotes the p-value is lower than 0.05. Results indicate the correlation is solid.

| Task | XNLI$^\dagger$ | XNLI$^\ddagger$ | PAWS-X$^\dagger$ | PAWS-X$^\ddagger$ |
|---|---|---|---|---|
| $\rho$ | 0.76* | 0.69* | 0.90* | 0.93* |

discrepancy. Besides, dissimilar languages often show differences in language characteristics (like vocabulary, word order), which also leads to the representation discrepancy. As a result, we assume that the cross-lingual transfer performance is closely related to the representation discrepancy between the source language and target languages.

Following Conneau et al. (2020b), we utilize the linear centered kernel alignment (CKA; Kornblith et al., 2019) score to indicate the cross-lingual representation discrepancy

$$\text{CKA}(X, Y) = \frac{||Y^\top X||_\text{F}^2}{||X^\top X||_\text{F}^2 ||Y^\top Y||_\text{F}^2}, \tag{1}$$

where X and Y are parallel sequences from the source and target languages, respectively. A higher CKA score denotes a smaller cross-lingual representation discrepancy.

To verify our assumption, we perform qualitative and quantitative analyses on the relationship between the CKA score and cross-lingual transfer performance. Figure 3 in Appendix B indicates a higher CKA score tends to induce better cross-lingual transfer performance. We also calculate the Spearman's rank correlation between the CKA score and the transfer performance in Table 2, which shows a strong correlation between them. Both the trend and correlation score confirm **the cross-lingual transfer performance is highly related to the cross-lingual representation discrepancy**.

## 4 METHODOLOGY: X-MIXUP

Based on the aforementioned analyses, we believe that reducing the cross-lingual representation discrepancy is the key to filling the cross-lingual transfer gap. In this section, we propose X-MIXUP to explicitly reduce the representation discrepancy by implementing the manifold mixup between the source language and target language. With X-MIXUP, the model can adaptively calibrate the representation discrepancy and give compromised representations for target languages. This section will first introduce the overall architecture of X-MIXUP and its details. After that, the training objectives and inference process will be shown.

### 4.1 OVERALL ARCHITECTURE

Figure 2 illustrates the overall architecture of X-Mixup. Sequences from the source and target languages are first encoded separately. Then within the encoder, X-MIXUP implements the manifold mixup between the paired sequences (original sequence and its translation) within a specific layer, where *Mixup Ratio* controls the degree of mixup and *Scheduled Sampling* schedules the data sampling process during training.

**Notations** We use $S$ to denote the source language and $T$ to denote the target language. $\boldsymbol{h}^l$ denotes the hidden states of a sequence in layer $l$. $\mathcal{D}$ denotes the real text data collection and $\tilde{\mathcal{D}}$ denotes the translation data collection. For downstream understanding tasks, there are annotation data in the source language $\mathcal{D}_S^\text{Train} = (\mathcal{X}_S^\text{Train}, \mathcal{Y}_S^\text{Train})$ and raw test data in the target language $\mathcal{D}_T^\text{Test} = (\mathcal{X}_T^\text{Test})$. Through translate-train, we can get pseudo-training data in the target language $\tilde{\mathcal{D}}_T^\text{Train} = (\tilde{\mathcal{X}}_T^\text{Train}, \tilde{\mathcal{Y}}_T^\text{Train})$. Similarly, through translate-test, we can get pseudo-test data in the source language $\tilde{\mathcal{D}}_S^\text{Test} = (\tilde{\mathcal{X}}_S^\text{Test})$. During training, the Scheduled Sampling process uses translation data[2] $\tilde{\mathcal{D}}_S^\text{Train} = (\tilde{\mathcal{X}}_S^\text{Train})$ from the source language. Note that we use translation data ($\tilde{\mathcal{X}}_T^\text{Train}$ and $\tilde{\mathcal{X}}_S^\text{Test}$) and translate-train labels ($\tilde{\mathcal{Y}}_T^\text{Train}$) from the official XTREME repository, which is in the same setting as baselines.

---

[2]These data are acquired by forward translation (from $S$ to $T$) then backward translation (from $T$ to $S$).

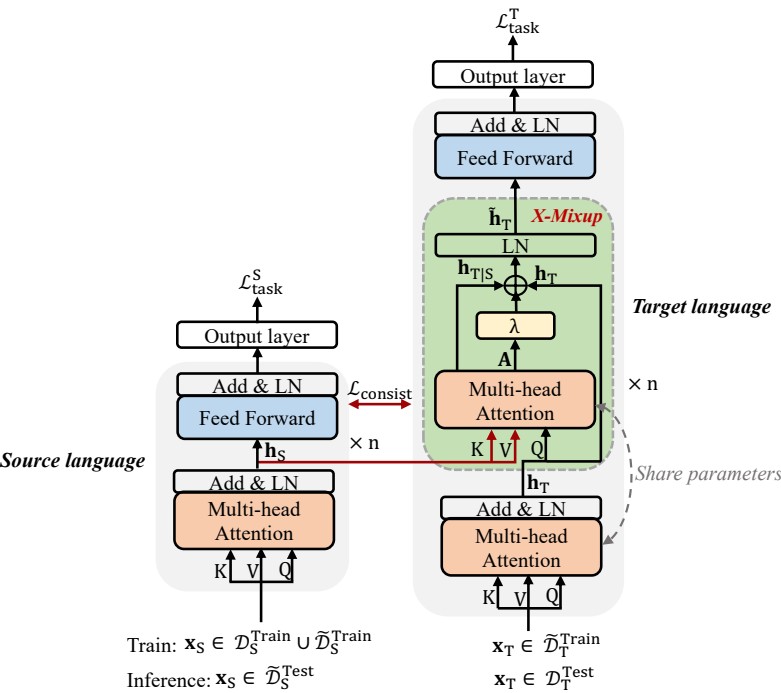

Figure 2: The model architecture of X-MIXUP, where the cross-lingual manifold mixup process is in the green block. Note that the manifold mixup process is implemented only in a certain layer (the same layer of both sides), and in other layers the process is omitted.

**Basic Model**   We use mBERT (Devlin et al., 2019) or XLM-R (Conneau et al., 2020a) as the backbone model. Within each layer, there are two sub-layers: the multi-head attention layer and the feed-forward layer[3], followed by the residual connection and layer norm. We use the same multi-head attention layer (see details in Appendix A.1) as BERT (Devlin et al., 2019), where inputs are query, key, and value respectively. In layer $l + 1$, the hidden states of the source sequence $\boldsymbol{x}_S$ and target sequence $\boldsymbol{x}_T$ are acquired by the multi-head attention

$$\boldsymbol{h}_S^{l+1} = \text{MultiHead}(\boldsymbol{h}_S^l, \boldsymbol{h}_S^l, \boldsymbol{h}_S^l), \ \boldsymbol{h}_T^{l+1} = \text{MultiHead}(\boldsymbol{h}_T^l, \boldsymbol{h}_T^l, \boldsymbol{h}_T^l). \tag{2}$$

**Manifold Mixup**   To reduce the cross-lingual representation discrepancy, a straightforward idea is to find compromised representations between the source and target languages. It's difficult to find such representations because of varying degrees of differences across languages, like vocabulary and word order. However, manifold mixup provides an elegant way to get intermediate representations by conducting linear interpolation on hidden states.

To extract target-related information from the source hidden states, the target hidden states are used as the query, and the source hidden states are used as the key and value. This cross-attention process is computed as

$$\boldsymbol{h}_{T|S}^{l+1} = \text{MultiHead}(\boldsymbol{h}_T^{l+1}, \boldsymbol{h}_S^{l+1}, \boldsymbol{h}_S^{l+1}), \tag{3}$$

which shares parameters with the multi-head attention. The manifold mixup process mixes the target hidden states $\boldsymbol{h}_T^{l+1}$ and the source-aware target hidden states $\boldsymbol{h}_{T|S}^{l+1}$ based on the *mixup ratio* $\lambda$

$$\tilde{\boldsymbol{h}}_T^{l+1} = \text{LN}(\lambda \boldsymbol{h}_{T|S}^{l+1} + (1 - \lambda) \boldsymbol{h}_T^{l+1}), \tag{4}$$

where $\lambda$ is an instance-level parameter, ranging from 0 to 1, and indicates the degree of manifold mixup. LN denotes the layer norm operation.

**Mixup Ratio**   The machine translation process may change the original semantics and introduce data noises in varying degrees (Castilho et al., 2017; Fomicheva et al., 2020). Thus we introduce the

---

[3]In this section, the feed-forward layer is omitted for simplification.

translation quality modeling in the mixup process to handle this problem. Following Fomicheva et al. (2020), we use the entropy of attention weights to measure the translation quality

$$\mathrm{H}(\boldsymbol{A}) = -\frac{1}{I}\sum_i^I \sum_j^J \boldsymbol{A}_{ji}\log\boldsymbol{A}_{ji}, \text{ where } \boldsymbol{A}_{ij} = \mathrm{softmax}(\frac{\boldsymbol{h}_{T_i}\boldsymbol{h}_{S_j}^\top}{\sqrt{n}}). \tag{5}$$

$I$ is the number of target tokens and $J$ is the number of source tokens. Lower entropy implies better cross-lingual alignment and higher translation quality.

To introduce the translation quality modeling into the manifold mixup process, we compute the mixup ratio as $\lambda = \lambda_0 \cdot \sigma[(\mathrm{H}(\boldsymbol{A}) + \mathrm{H}(\boldsymbol{A}^\top))W + b]$, where $\sigma$ is the sigmoid function, and $W$, $b$ are trainable parameters. $\lambda_0$ is the max value of the mixup ratio, which is set to 0.5 in this paper. We consider two-way alignment in the translation quality modeling, i.e. $\mathrm{H}(\boldsymbol{A})$ and $\mathrm{H}(\boldsymbol{A}^\top)$.

**Scheduled Sampling** The source sequences utilized in training and inference are drawn from different distributions. During training, the source sequence is a real text from $\mathcal{D}_S^{\mathrm{Train}}$, while during inference, the source sequence is a translation from $\tilde{\mathcal{D}}_S^{\mathrm{Test}}$. This discrepancy, commonly called the exposure bias, leads to a gap between training and inference.

Motivated by the scheduled sampling approach (Bengio et al., 2015) in NMT, we sample the source sequence dynamically during training. Specifically, the source sequence fed into the manifold mixup is either a real text from $\mathcal{D}_S^{\mathrm{Train}}$ or translation from $\tilde{\mathcal{D}}_S^{\mathrm{Train}}$ with a certain probability $p$

$$\begin{cases} p <= p^*, & \boldsymbol{x}_s \in \mathcal{D}_S^{\mathrm{Train}}, \\ p > p^*, & \boldsymbol{x}_s \in \tilde{\mathcal{D}}_S^{\mathrm{Train}}, \end{cases} \tag{6}$$

where $p^*$ is decreasing during training to match the situation of inference. We utilize the inverse sigmoid decay (Bengio et al., 2015), which decreases $p^*$ as a function of the index of mini-batch.

### 4.2 FINAL TRAINING OBJECTIVE

The training loss is composed of two parts: the task loss and the consistency loss

$$\mathcal{L} = \underbrace{\mathcal{L}_{\mathrm{task}}}_{\text{task loss}} + \underbrace{\mathrm{MSE}(\boldsymbol{r}_S, \boldsymbol{r}_T) + \mathrm{KL}(\boldsymbol{p}_S, \boldsymbol{p}_T)}_{\text{consistency loss}}. \tag{7}$$

where $\mathrm{MSE}(\cdot)$ is Mean Squared Error and $\mathrm{KL}(\cdot)$ is Kullback-Leibler divergence. $\boldsymbol{r}_*$ is the sequence representation[4] and $\boldsymbol{p}_*$ is the predicted probability distribution of downstream tasks.

The task loss $\mathcal{L}_{\mathrm{task}}$ is the sum of the source language task loss $\mathcal{L}_{\mathrm{task}}^S$ and target language one $\mathcal{L}_{\mathrm{task}}^T$, weighted by the hyper-parameter $\alpha$, which is utilized to balance the training process

$$\mathcal{L}_{\mathrm{task}} = \alpha\mathcal{L}_{\mathrm{task}}^S + (1 - \alpha)\mathcal{L}_{\mathrm{task}}^T. \tag{8}$$

For classification, structured prediction, and span extraction tasks, the task loss is the cross-entropy loss (see details in Appendix A.2). For structured prediction tasks, it is non-trivial to implement the token-level label mapping across different languages. Thus we use the label probability distribution, predicted by the source language task model, as the pseudo-label for training, where tokens and labels are corresponding.

The consistency loss is composed of two parts: the representation consistency loss and the prediction consistency loss. The first loss is a regularization term and provides a way to align representations across different languages (Ruder et al., 2019). The second loss is to make better use of the supervision of downstream tasks. It only exists in the classification task, as the translation process does not change the label of this task, while in other tasks, it does.

### 4.3 INFERENCE

During inference, the manifold mixup process is the same as training, except for the Scheduled Sampling process. Concretely, for the source language, only translation data are available in the

---

[4]We utilize the mean pooling of the last layer's hidden states as the sequence representation, which is independent of the sequence length.

Table 3: Main results on the XTREME benchmark. † denotes using other data augmentation strategy in addition to machine translation. ‡ denotes results from Ruder et al. (2021), which is an updated version of Hu et al. (2020).

| Model | Pair Sentence | | Structured Prediction | | Question Answering | | | Avg. |
| --- | --- | --- | --- | --- | --- | --- | --- | --- |
| | XNLI | PAWS-X | POS | NER | XQuAD | MLQA | TyDiQA | |
| Metrics | Acc | Acc | F1 | F1 | F1/EM | F1/EM | F1/EM | - |
| *Based on XLM-R-large* | | | | | | | | |
| XLM-R (Hu et al., 2020) | 79.2 | 86.4 | 73.8 | 65.4 | 76.6/60.8 | 71.6/53.2 | 65.1/45.0 | 70.1 |
| Trans-train (Wei et al., 2020) | 82.9 | 90.1 | 74.6 | 66.8 | 80.4/65.6 | 72.4/54.7 | 66.2/48.2 | 72.6 |
| Filter (Fang et al., 2020) | 83.9 | 91.4 | 76.2 | 67.7 | 82.4/68.0 | 76.2/57.7 | 68.3/50.9 | 74.4 |
| xTUNE (Zheng et al., 2021) | 84.8 | 91.6 | **79.3**† | **69.9**† | 82.5/69.0† | 75.0/57.1† | **75.4/60.8**† | **76.5** |
| X-MIXUP | **85.3** | **91.8** | 78.4 | 69.0 | **82.6/69.3** | **76.5/58.1** | 69.0/52.8 | 75.5 |
| *Based on mBERT* | | | | | | | | |
| mBERT (Hu et al., 2020) | 65.4 | 81.9 | 71.5 | 62.2 | 64.5/49.4 | 61.4/44.2 | 59.7/43.9 | 63.2 |
| Joint-Align (Zhao et al., 2020) | 72.3 | - | - | - | - | - | - | - |
| Trans-train (Hu et al., 2020) | 75.1 | 88.9 | - | - | 72.4/58.3 | 67.6/49.8 | 59.5/45.8‡ | - |
| X-MIXUP | **78.8** | **89.7** | **76.5** | **65.0** | **73.3/58.9** | **69.0/50.9** | **60.8/46.5** | **70.0** |

inference stage, without real data, so we use $x_s \in \tilde{\mathcal{D}}_S^{\text{Test}}$. For classification tasks, we synthesize the predictions of both the source and target sequences by taking the mean of the predicted probability distributions as the final prediction. For structured prediction and QA tasks, we only consider the prediction of the target sequence.

## 5 EXPERIMENTS

This section first introduces the cross-lingual understanding benchmark, XTREME. Then briefly introduces the configurations of downstream tasks and baselines. Finally, shows the main results of baselines and X-MIXUP on XTREME.

### 5.1 TASKS AND SETTINGS

**Tasks** In our experiments, we focus on three types of tasks in XTREME: (1) sentence pair classification task: XNLI (Conneau et al., 2018) and PAWS-X (Yang et al., 2019); (2) structured prediction task: POS (Nivre et al., 2018) and NER (Pan et al., 2017); (3) question answering task: XQuAD (Artetxe et al., 2020), MLQA (Lewis et al., 2020) and TyDiQA (Clark et al., 2020). The details of these datasets can refer to Hu et al. (2020). We utilize the translate-train and translate-test data from the XTREME repo[5], which also provide the pseudo-label of translate-train data for classification tasks and question answering tasks. The rest translation data are from Google Translate[6].

**Models** Experiments are based on two multilingual pre-trained models: mBERT and XLM-R. We use the pre-trained models of Huggingface Transformers[7] as the backbone model.

**Hyper-parameters** We select XNLI, POS, and MLQA as representative tasks to search for the best hyper-parameters. The final model is selected based on the averaged performance of all languages on the dev set. We perform grid search over the balance training parameter $\alpha$ and learning rate from [0.2, 0.4, 0.6, 0.8] and [3e-6, 5e-6, 2e-5, 3e-5]. We also search for the best manifold mixup layer from [1, 4, 8, 12, 16, 20, 24]. In final results, we implement mixup in the first layer for classification tasks, 4th layer for structured prediction tasks. For QA tasks, we implement mixup in the 16th layer for large model, 8th layer for base model. Concrete details of experiments are presented in Appendix C.1.

### 5.2 BASELINES

We conduct experiments on two strong multilingual pre-trained models to verify the generality of methods: (1) **mBERT** Multilingual BERT is a 12-layer transformer model pre-trained on the

---

[5] https://github.com/google-research/xtreme.

[6] https://translate.google.com/.

[7] We use bert-base-multilingual-cased for mBERT and xlm-roberta-large for XLM-R.

Table 4: Comparisons between X-MIXUP and XTUNE under the same setting: XLM-R-base model and machine translation data augmentation. Results of XTUNE are from Zheng et al. (2021) Table 4.

| Model | XNLI | POS | MLQA |
|---|---|---|---|
| XTUNE$_{\mathcal{R}_1}$ (Zheng et al., 2021) | 79.7 | - | - |
| XTUNE$_{\mathcal{R}_2}$ (Zheng et al., 2021) | 78.9 | 76.6 | 68.7/51.1 |
| X-MIXUP | **80.4** | **77.8** | **71.2/53.1** |

Wikipedias of 104 languages. (2) **XLM-R** XLM-R-large is a 24-layer transformer model pre-trained on 2.5T data extracted from Common Crawl covering 100 languages. Based on them, these are some strong baselines: (1) **Trans-train** Abbreviation for Translate-train. The training set of the source language is machine-translated to each target language and then the model is trained on the concatenation of all training sets. (2) **Joint-Align** Zhao et al. (2020) aligns the monolingual sub-spaces of the source and target language by minimizing the distances of embeddings for matched word pairs. (3) **Filter** Fang et al. (2020) splices the representation of the target sequence and its translation in intermediate layers to extract multilingual knowledge. (4) **XTUNE** Zheng et al. (2021) uses two types of consistency regularization based on four types of data augmentation.

## 5.3 MAIN RESULTS

Results on the XTREME benchmark are shown in Table 3. Concrete results for each task are presented in Appendix C.2. Compared with strong baselines, X-MIXUP shows its superiority across different backbones and tasks, which indicates its generality. X-MIXUP outperforms Trans-train by 2.2% based on mBERT, and X-MIXUP outperforms Filter by 1.5% based on XLM-R. The superiority of X-MIXUP over Filter is that X-MIXUP gives a calibrated representation for target languages, not just the concatenation of two representations. Besides, X-MIXUP considers the noise of translation data and limits the noise propagation by introducing mixup ratio.

XTUNE achieves the best results on structured prediction tasks and the low-resource QA task TyDiQA (only 3.7k training data in English), but XTUNE uses three other data augmentation approaches in addition to machine translation. To make a fairer comparison, we conduct experiments under the same setting in Table 4, which indicates X-MIXUP outperforms XTUNE on three types of tasks with only machine translation data augmentation. Besides, X-MIXUP only needs one-stage training, while XTUNE implements a two-stage training algorithm. However, X-MIXUP and XTUNE are complementary, where the former focuses on finding better representations for target languages while the latter concentrates on the cross-lingual data augmentation and consistency regularization.

## 6 ANALYSIS AND DISCUSSION

To better understand X-MIXUP and explore how X-MIXUP influences the cross-lingual transfer performance, we conduct analyses[8] on several questions. Results show X-MIXUP achieves performance improvements across different languages and it also reduces the cross-lingual representation discrepancy obviously. Table 6 in Appendix B verifies the effectiveness of X-Mixup on both seen and unseen languages. Besides, ablation results show the cross-lingual manifold mixup training contributes a lot to cross-lingual transfer.

**(Q1)** *How X-MIXUP influences the cross-lingual representation discrepancy?* Language centroid (Rosenberg & Hirschberg, 2007) is the mean of the representations within each language. We plot the language centroid of different methods (see Figure 4 in Appendix B), which indicates X-MIXUP brings closer language centroids significantly. We also calculate the CKA scores of the XNLI dataset (see Table 7 in Appendix B). Results show X-MIXUP reduces the cross-lingual representation discrepancy evenly across different target languages, improving the CKA score by 10.4% on average. In conclusion, both the language centroids visualization and the CKA score improvement indicate X-MIXUP reduces the cross-lingual representation discrepancy effectively.

---

[8]In this section, we utilize the XLM-R-large model as the backbone model.

Table 5: Ablation results on X-MIXUP, where w/o mixup denotes remove the cross-lingual manifold mixup during training and inference and $\lambda = \lambda_0$ denotes a constant mixup ratio.

| Model | XNLI | POS | MLQA |
|---|---|---|---|
| X-MIXUP | 85.3 | 78.4 | 76.5/58.1 |
|    w/o mixup | 82.9 | 75.7 | 72.7/54.8 |
|    w/o mixup inference | 84.0 | 77.6 | 75.6/57.3 |
|    w/o scheduled sampling | 84.6 | 78.0 | 76.3/57.9 |
|    w/o consistency loss | 84.2 | 78.0 | 76.5/58.0 |
|    $\lambda = \lambda_0$ | 84.1 | 77.8 | 75.8/57.5 |

**(Q2)** *How* **X-MIXUP** *influences the cross-lingual transfer gap?* We compare the cross-lingual transfer gap in Appendix B Table 8. Compared with Trans-train, X-MIXUP reduces the averaged gap by 39.8% and shows its superiority across three types of tasks. Compared with state-of-the-art methods, X-MIXUP achieves the smallest cross-lingual transfer gap on four out of seven datasets, which suggests the effectiveness of X-MIXUP on classification and QA tasks.

**(Q3)** *What is the essential component of* **X-MIXUP***?* There are five major components of X-MIXUP: cross-lingual manifold mixup training, mixup inference, Mixup Ratio, Scheduled Sampling, and consistency loss. To better understand X-MIXUP, we implement ablation studies in Table 5. Comparisons between X-MIXUP and w/o mixup show the effectiveness of cross-lingual manifold mixup across different tasks, and even without mixup inference (translate-test data), the mixup training can also achieve 2.6% improvements on average. Besides, comparisons between X-MIXUP and $\lambda = \lambda_0$ show the effectiveness of introducing the translation quality modeling in the mixup process. Scheduled sampling achieves more performance improvements on the classification task, as the task shares labels across languages, and scheduled sampling can prevent the model from solely relying on the gold source sequence to make predictions. In addition, the consistency loss is also more effective on the classification task, because there is additional prediction consistency loss which can transfer the task capability from the source language to target languages. Detailed ablation results on the consistency loss are shown in Appendix B Table 9, which shows the KL consistency loss contributes more than the MSE consistency loss on the classification task.

**(Q4)** *Which layer is the best to implement the manifold mixup?* We implement the cross-lingual manifold mixup in different layers (see Figure 5 in Appendix B for details) and find different tasks prefer different mixup layers. Although different tasks have their own preferences, no matter which layer we mix, the cross-lingual transfer performance can be improved, except for mixing within a higher layer on classification tasks. The drop in classification task is mainly because the source and target sequences share the same task label. Performing mixup in a higher layer may make the model rely on the source sequence and ignore the target sequence. The structured prediction task is not sensitive to the mixup layer, mainly because this task relies on both the short and long dependence. For QA tasks, the cross-lingual transfer performance shows a trend from rise to decline as the mixup layer increases. The QA task needs higher-level understanding, but higher layers are more language-specific, where sequences from different languages have different gold answers.

## 7 CONCLUSION

This paper focuses on enhancing the cross-lingual transfer performance on understanding tasks. Considering the large cross-lingual transfer gap in recent works, this paper first analyses related factors and finds this gap is strongly associated with the cross-lingual representation discrepancy. Then X-MIXUP is proposed to alleviate the discrepancy, which gives compromised representations for target languages by implementing the manifold mixup between the source and target languages. Empirical evaluations on XTREME verify the effectiveness of X-MIXUP across different tasks and languages. Besides, both the visualization and quantitative analyses show X-MIXUP reduces the cross-lingual representation discrepancy effectively. Furthermore, X-Mixup can also be applied to the multilingual pre-training process by implementing the cross-lingual manifold mixup on parallel data. Findings on the relationship between the cross-lingual transfer performance and representation discrepancy shed light on a promising way to boost cross-lingual transfer for future research.

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

# A    METHOD DETAILS

## A.1    MULTI-HEAD ATTENTION

In the multi-head attention layer, multiple attention heads are concatenated

$$\text{MultiHead}(Q, K, V) = \text{Concat}(\text{head}_{1,...,h})\boldsymbol{W}^O, \tag{9}$$

and each head is the scaled dot-product attention

$$\text{head}_i = \text{Attention}(Q\boldsymbol{W}_i^q, K\boldsymbol{W}_i^k, V\boldsymbol{W}_i^v), \tag{10}$$

$$\text{Attention}(Q, K, V) = \text{softmax}(\frac{QK^\top}{\sqrt{d}})V, \tag{11}$$

where $\boldsymbol{W}^O$, $\boldsymbol{W}^q$, $\boldsymbol{W}^k$ and $\boldsymbol{W}^v$ are trainable parameters.

## A.2    TRAINING OBJECTIVE

For classification tasks (e.g. NLI), the task loss is the cross-entropy loss

$$\mathcal{L}_{\text{task}} = -\sum_{j}^{C} \boldsymbol{y}_j \log \boldsymbol{p}_j, \tag{12}$$

where $C$ is the size of the label set.

For structured prediction tasks (e.g. POS) and span extraction tasks (e.g. QA), the task loss is also the cross-entropy loss

$$\mathcal{L}_{\text{task}} = -\sum_{i}^{n} \sum_{j}^{C} \boldsymbol{y}_{ij} \log \boldsymbol{p}_{ij}. \tag{13}$$

where $n$ is the sequence length.

## B  ANALYSIS RESULTS

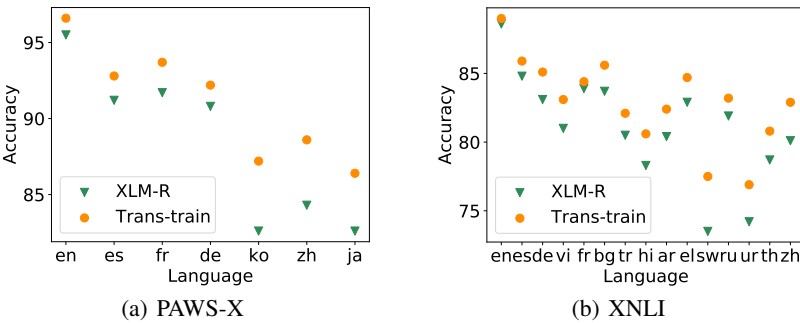

(a) PAWS-X

(b) XNLI

Figure 3: Performances on PAWS-X and XNLI test set, where languages are sorted by decreasing CKA scores. The trend indicates the performance gets worse along with the CKA score decreasing.

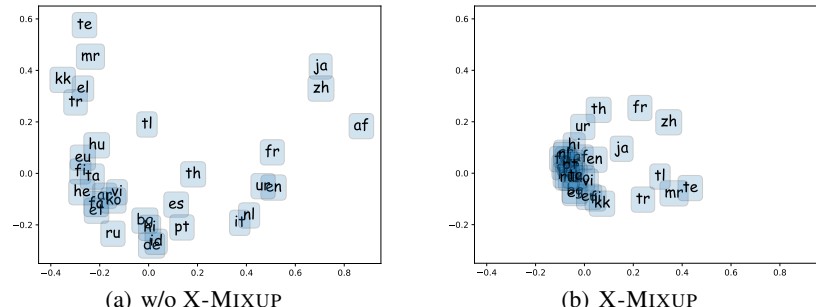

(a) w/o X-MIXUP

(b) X-MIXUP

Figure 4: Language centroids visualization of the POS test set, which indicates X-Mixup brings closer these centroids obviously.

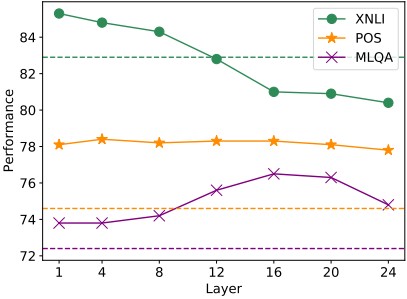

Figure 5: Performances on implementing X-MIXUP (solid line) in different layers and Trans-train (dashed line) on three downstream tasks.

Table 6: Performances on XNLI test set, where Trans-train and X-MIXUP are trained on 8 seen languages and tested on both these seen languages and 7 unseen languages. $\Delta$ is the performance difference between X-MIXUP and Trans-train. Results show X-MIXUP performs better than Trans-train by a large margin on both seen and unseen languages.

| Model | Seen Languages | | | | | | | | Unseen Languages | | | | | | | Avg. |
|---|---|---|---|---|---|---|---|---|---|---|---|---|---|---|---|---|
| | en | bg | el | fr | ru | th | ur | zh | ar | de | es | hi | sw | tr | vi | |
| Trans-train | 87.6 | 84.7 | 84.2 | 84.6 | 82.9 | 80.1 | 76.5 | 83.0 | 82.6 | 84.5 | 85.0 | 80.6 | 77.8 | 82.7 | 82.7 | 82.6 |
| X-MIXUP | 89.5 | 87.1 | 86.3 | 86.8 | 84.7 | 82.7 | 79.0 | 85.0 | 85.3 | 86.3 | 86.9 | 82.9 | 80.3 | 84.5 | 84.5 | 84.8 |
| $\Delta$ | +1.9 | +2.4 | +1.9 | +2.2 | +1.8 | +2.6 | +2.5 | +2.0 | +2.7 | +1.8 | +1.9 | +1.7 | +2.5 | +1.8 | +1.8 | +1.8 |

Table 7: CKA scores and performances on XNLI test set, where $\Delta$ is the score or performance difference between X-MIXUP and Trans-train. Results show X-MIXUP improves the CKA scores evenly across different target languages and the performance improvements are diverse. There is no obvious correlation between the CKA score improvement and performance improvement.

| Model | en | es | de | vi | fr | bg | tr | el | ru | ar | hi | sw | ur | th | zh | Avg. |
|---|---|---|---|---|---|---|---|---|---|---|---|---|---|---|---|---|
| *CKA score* | | | | | | | | | | | | | | | | |
| Trans-train | 1.00 | 0.81 | 0.80 | 0.78 | 0.78 | 0.78 | 0.76 | 0.75 | 0.75 | 0.74 | 0.74 | 0.73 | 0.72 | 0.72 | 0.72 | 0.77 |
| X-MIXUP | 1.00 | 0.88 | 0.87 | 0.86 | 0.86 | 0.86 | 0.85 | 0.83 | 0.82 | 0.84 | 0.83 | 0.82 | 0.81 | 0.80 | 0.79 | 0.85 |
| $\Delta$ | 0.00 | 0.07 | 0.07 | 0.08 | 0.08 | 0.08 | 0.09 | 0.08 | 0.07 | 0.10 | 0.09 | 0.09 | 0.09 | 0.08 | 0.07 | 0.08 |
| *Performance* | | | | | | | | | | | | | | | | |
| Trans-train | 88.6 | 85.7 | 84.5 | 82.6 | 84.2 | 85.2 | 82.1 | 84.5 | 81.8 | 82.2 | 80.8 | 77.0 | 77.7 | 80.2 | 82.7 | 82.6 |
| X-MIXUP | 89.9 | 87.7 | 86.9 | 85.4 | 87.1 | 87.3 | 84.9 | 86.8 | 85.1 | 85.2 | 83.5 | 81.2 | 79.6 | 83.2 | 85.2 | 85.3 |
| $\Delta$ | +1.3 | +2.0 | +2.4 | +2.8 | +2.9 | +2.1 | +2.8 | +2.3 | +3.3 | +3.0 | +2.7 | +4.2 | +1.9 | +3.0 | +2.5 | +2.7 |

Table 8: The cross-lingual transfer gap (lower is better) of different methods on the XTREME benchmark. For QA tasks, we only show EM scores. [†] denotes results from Wei et al. (2020). Overall, X-MIXUP achieves the smallest cross-lingual transfer gap on four out of seven datasets.

| Model | XNLI | PAWS-X | POS | NER | XQuAD | MLQA | TyDiQA | Avg. |
|---|---|---|---|---|---|---|---|---|
| mBERT (Hu et al., 2020) | 16.5 | 14.1 | 25.5 | 23.6 | 25.0 | 27.5 | 22.2 | 22.1 |
| XLM-R (Hu et al., 2020) | 10.2 | 12.4 | 24.3 | 19.8 | 16.3 | 19.1 | 13.1 | 16.5 |
| Trans-train (Hu et al., 2020) | 7.3 | 9.0 | 22.4[†] | 20.5[†] | 17.6 | 22.2 | 24.2 | 17.6 |
| Filter (Fang et al., 2020) | 6.0 | **5.2** | 19.7 | 16.3 | 7.3 | 15.7 | 9.2 | 11.3 |
| xTUNE (Zheng et al., 2021) | 5.5 | **5.2** | **17.3** | **14.8** | 10.1 | 18.5 | **0.9** | **10.3** |
| X-MIXUP | **4.9** | **5.2** | 18.1 | 15.9 | **6.7** | **13.9** | 9.6 | 10.6 |

Table 9: Ablation results on the consistency loss, which show the KL consistency loss contributes more than the MSE consistency loss on the classification task.

| Model | XNLI | POS | MLQA |
|---|---|---|---|
| X-MIXUP | 85.3 | 78.4 | 76.5/58.1 |
| w/o MSE consistency loss | 84.6 | 78.0 | 76.5/58.0 |
| w/o KL consistency loss | 84.3 | - | - |
| w/o both | 84.2 | - | - |

# C  EXPERIMENTAL DETAILS

## C.1  HYPER-PARAMETERS

For all tasks, we fine-tune on 8 Nvidia V100-32GB GPU cards with the batch size 64. For XQuAD and MLQA, we finetune 2 epochs. For other tasks, we finetune 4 epochs. There is no dev set in XQuAD, so we use the dev set of MLQA for the model selection. Table 10 shows hyper-parameters used for X-MIXUP.

Table 10: Hyper-parameters used for X-MIXUP, where $\alpha$ is used for balanced training in Eq 8 and $p_k$ is the scheduled sampling decay rate.

| Parameter | Classification | Structured Prediction | QA |
|---|---|---|---|
| $\alpha$ | 0.4 | 0.8 | 0.2 |
| $p_k$ | 1000 | 1000 | 2000 |

## C.2  DETAILED RESULTS

Detailed results of each tasks and languages are shown below. Results of mBERT, XLM, MMTE and XLM-R are from XTREME (Hu et al., 2020). Results of Filter is the best results of Fang et al. (2020).

| Model | en | ar | bg | de | el | es | fr | hi | ru | sw | th | tr | ur | vi | zh | Avg. |
|---|---|---|---|---|---|---|---|---|---|---|---|---|---|---|---|---|
| mBERT | 81.9 | 73.8 | 77.6 | 77.6 | 75.9 | 79.1 | 77.8 | 70.7 | 75.4 | 70.5 | 70.0 | 74.3 | 67.4 | 77.0 | 77.6 | 75.1 |
| XLM | 82.8 | 66.0 | 71.9 | 72.7 | 70.4 | 75.5 | 74.3 | 62.5 | 69.9 | 58.1 | 65.5 | 66.4 | 59.8 | 70.7 | 70.2 | 69.1 |
| MMTE | 79.6 | 64.9 | 70.4 | 68.2 | 67.3 | 71.6 | 69.5 | 63.5 | 66.2 | 61.9 | 66.2 | 63.6 | 60.0 | 69.7 | 69.2 | 67.5 |
| XLM-R | 88.6 | 82.2 | 85.2 | 84.5 | 84.5 | 85.7 | 84.2 | 80.8 | 81.8 | 77.0 | 80.2 | 82.1 | 77.7 | 82.6 | 82.7 | 82.6 |
| Filter | 89.5 | 83.6 | 86.4 | 85.6 | 85.4 | 86.6 | 85.7 | 81.1 | 83.7 | 78.7 | 81.7 | 83.2 | 79.1 | 83.9 | 83.8 | 83.9 |
| XTUNE | 89.9 | 84.0 | 87.0 | 86.5 | 86.2 | 87.4 | 86.6 | 83.2 | 85.2 | 80.0 | 82.7 | 84.1 | 79.6 | 84.8 | 84.3 | 84.8 |
| X-MIXUP | 89.9 | 85.2 | 87.3 | 86.9 | 86.8 | 87.7 | 87.1 | 83.5 | 85.1 | 81.2 | 83.2 | 84.9 | 79.6 | 85.4 | 85.2 | 85.3 |

Table 11: XNLI accuracy scores for each language.

| Model | en | de | es | fr | ja | ko | zh | Avg. |
|---|---|---|---|---|---|---|---|---|
| mBERT | 94.0 | 85.7 | 87.4 | 87.0 | 73.0 | 69.6 | 77.0 | 81.9 |
| XLM | 94.0 | 85.9 | 88.3 | 87.4 | 69.3 | 64.8 | 76.5 | 80.9 |
| MMTE | 93.1 | 85.1 | 87.2 | 86.9 | 72.0 | 69.2 | 75.9 | 81.3 |
| XLM-R | 94.7 | 89.7 | 90.1 | 90.4 | 78.7 | 79.0 | 82.3 | 86.4 |
| Filter | 95.9 | 92.8 | 93.0 | 93.7 | 87.4 | 87.6 | 89.6 | 91.5 |
| XTUNE | 96.1 | 92.6 | 93.1 | 93.9 | 87.8 | 89.0 | 88.8 | 91.6 |
| X-MIXUP | 96.3 | 93.2 | 93.6 | 94.6 | 87.3 | 88.2 | 89.5 | 91.8 |

Table 12: PAWS-X accuracy scores for each language.

| Model | af | ar | bg | de | el | en | es | et | eu | fa | fi | fr | he | hi | hu | id | it |
|---|---|---|---|---|---|---|---|---|---|---|---|---|---|---|---|---|---|
| mBERT | 86.6 | 56.2 | 85.0 | 85.2 | 81.1 | 95.5 | 86.9 | 79.1 | 60.7 | 66.7 | 78.9 | 84.2 | 56.2 | 67.2 | 78.3 | 71.0 | 88.4 |
| XLM | 88.5 | 63.1 | 85.0 | 85.8 | 84.3 | 95.4 | 85.8 | 78.3 | 62.8 | 64.7 | 78.4 | 82.8 | 65.9 | 66.2 | 77.3 | 70.2 | 87.4 |
| XLM-R | 89.8 | 67.5 | 88.1 | 88.5 | 86.3 | 96.1 | 88.3 | 86.5 | 72.5 | 70.6 | 85.8 | 87.2 | 68.3 | 76.4 | 82.6 | 72.4 | 89.4 |
| Filter | 88.7 | 66.1 | 88.5 | 89.2 | 88.3 | 96.0 | 89.1 | 86.3 | 78.0 | 70.8 | 86.1 | 88.9 | 64.9 | 76.7 | 82.6 | 72.6 | 89.8 |
| xTune | 90.4 | 72.8 | 89.0 | 89.4 | 87.0 | 96.1 | 88.8 | 88.1 | 73.1 | 74.7 | 87.2 | 89.5 | 83.5 | 77.7 | 83.6 | 73.2 | 90.5 |
| X-Mixup | 89.4 | 70.1 | 88.8 | 88.7 | 86.7 | 96.0 | 89.0 | 88.3 | 76.2 | 72.5 | 87.0 | 88.2 | 82.4 | 78.0 | 83.8 | 72.4 | 90.3 |

| Model | ja | kk | ko | mr | nl | pt | ru | ta | te | th | tl | tr | ur | vi | yo | zh | Avg. |
|---|---|---|---|---|---|---|---|---|---|---|---|---|---|---|---|---|---|
| mBERT | 49.2 | 70.5 | 49.6 | 69.4 | 88.6 | 86.2 | 85.5 | 59.0 | 75.9 | 41.7 | 81.4 | 68.5 | 57.0 | 53.2 | 55.7 | 61.6 | 71.5 |
| XLM | 49.0 | 70.2 | 50.1 | 68.7 | 88.1 | 84.9 | 86.5 | 59.8 | 76.8 | 55.2 | 76.3 | 66.4 | 61.2 | 52.4 | 20.5 | 65.4 | 71.3 |
| XLM-R | 15.9 | 78.1 | 53.9 | 80.8 | 89.5 | 87.6 | 89.5 | 65.2 | 86.6 | 47.2 | 92.2 | 76.3 | 70.3 | 56.8 | 24.6 | 25.7 | 73.8 |
| Filter | 40.4 | 80.4 | 53.3 | 86.4 | 89.4 | 88.3 | 90.5 | 65.3 | 87.3 | 57.2 | 94.1 | 77.0 | 70.9 | 58.0 | 43.1 | 53.1 | 76.9 |
| xTune | 65.3 | 79.8 | 56.0 | 85.5 | 89.7 | 89.3 | 90.8 | 65.7 | 85.5 | 61.4 | 93.8 | 78.3 | 74.0 | 57.5 | 27.9 | 68.8 | 79.3 |
| X-Mixup | 62.7 | 79.0 | 55.3 | 84.8 | 89.6 | 88.8 | 90.1 | 63.6 | 87.4 | 59.9 | 93.1 | 77.1 | 72.4 | 59.4 | 27.3 | 68.3 | 78.4 |

Table 13: POS results (F1) for each language.

| Model | en | af | ar | bg | bn | de | el | es | et | eu | fa | fi | fr | he | hi | hu | id | it | ja | jv |
|---|---|---|---|---|---|---|---|---|---|---|---|---|---|---|---|---|---|---|---|---|
| mBERT | 85.2 | 77.4 | 41.1 | 77.0 | 70.0 | 78.0 | 72.5 | 77.4 | 75.4 | 66.3 | 46.2 | 77.2 | 79.6 | 56.6 | 65.0 | 76.4 | 53.5 | 81.5 | 29.0 | 66.4 |
| XLM | 82.6 | 74.9 | 44.8 | 76.7 | 70.0 | 78.1 | 73.5 | 74.8 | 74.8 | 62.3 | 49.2 | 79.6 | 78.5 | 57.7 | 66.1 | 76.5 | 53.1 | 80.7 | 23.6 | 63.0 |
| MMTE | 77.9 | 74.9 | 41.8 | 75.1 | 64.9 | 71.9 | 68.3 | 71.8 | 74.9 | 62.6 | 45.6 | 75.2 | 73.9 | 54.2 | 66.2 | 73.8 | 47.9 | 74.1 | 31.2 | 63.9 |
| XLM-R | 84.7 | 78.9 | 53.0 | 81.4 | 78.8 | 78.8 | 79.5 | 79.6 | 79.1 | 60.9 | 61.9 | 79.2 | 80.5 | 56.8 | 73.0 | 79.8 | 53.0 | 81.3 | 23.2 | 62.5 |
| Filter | 83.5 | 80.4 | 60.7 | 83.5 | 78.4 | 80.4 | 80.7 | 74.0 | 81.0 | 66.9 | 71.3 | 80.2 | 79.9 | 57.4 | 74.3 | 82.2 | 54.0 | 81.9 | 24.3 | 63.5 |
| xTune | 85.0 | 80.4 | 59.1 | 84.8 | 79.1 | 80.5 | 82.0 | 78.1 | 81.5 | 64.5 | 65.9 | 82.2 | 81.9 | 62.0 | 75.0 | 82.8 | 55.8 | 83.1 | 30.5 | 65.9 |
| X-Mixup | 84.5 | 79.0 | 58.4 | 84.0 | 81.4 | 80.6 | 81.4 | 73.8 | 81.5 | 65.7 | 61.6 | 80.4 | 80.3 | 64.4 | 74.7 | 82.0 | 53.4 | 82.2 | 38.8 | 63.5 |

| Model | ka | kk | ko | ml | mr | ms | my | nl | pt | ru | sw | ta | te | th | tl | tr | ur | vi | yo | zh |
|---|---|---|---|---|---|---|---|---|---|---|---|---|---|---|---|---|---|---|---|---|
| mBERT | 64.6 | 45.8 | 59.6 | 52.3 | 58.2 | 72.7 | 45.2 | 81.8 | 80.8 | 64.0 | 67.5 | 50.7 | 48.5 | 3.6 | 71.7 | 71.8 | 36.9 | 71.8 | 44.9 | 42.7 |
| XLM | 67.7 | 57.2 | 26.3 | 59.4 | 62.4 | 69.6 | 47.6 | 81.2 | 77.9 | 63.5 | 68.4 | 53.6 | 49.6 | 0.3 | 78.6 | 71.0 | 43.0 | 70.1 | 26.5 | 32.4 |
| MMTE | 60.9 | 43.9 | 58.2 | 44.8 | 58.5 | 68.3 | 42.9 | 74.8 | 72.9 | 58.2 | 66.3 | 48.1 | 46.9 | 3.9 | 64.1 | 61.9 | 37.2 | 68.1 | 32.1 | 28.9 |
| XLMR | 71.6 | 56.2 | 60.0 | 67.8 | 68.1 | 57.1 | 54.3 | 84.0 | 81.9 | 69.1 | 70.5 | 59.5 | 55.8 | 1.3 | 73.2 | 76.1 | 56.4 | 79.4 | 33.6 | 33.1 |
| Filter | 71.0 | 51.1 | 63.8 | 70.2 | 69.8 | 69.3 | 59.0 | 84.6 | 82.1 | 71.1 | 70.6 | 64.3 | 58.7 | 2.4 | 74.4 | 83.0 | 73.4 | 75.8 | 42.9 | 35.4 |
| xTune | 76.3 | 56.9 | 67.1 | 72.6 | 71.5 | 72.5 | 66.7 | 85.8 | 82.1 | 75.2 | 72.4 | 66.0 | 61.8 | 1.1 | 77.5 | 83.7 | 75.6 | 80.8 | 44.9 | 36.5 |
| X-Mixup | 76.5 | 51.7 | 63.9 | 69.8 | 71.2 | 70.4 | 67.9 | 84.5 | 83.1 | 73.5 | 70.7 | 65.6 | 59.3 | 4.4 | 75.0 | 81.8 | 73.1 | 78.2 | 41.6 | 47.8 |

Table 14: NER results (F1) for each language.

| Model | en | ar | de | el | es | hi | ru | th | tr | vi | zh | Avg. |
|---|---|---|---|---|---|---|---|---|---|---|---|---|
| mBERT | 83.5 / 72.2 | 61.5 / 45.1 | 70.6 / 54.0 | 62.6 / 44.9 | 75.5 / 56.9 | 59.2 / 46.0 | 71.3 / 53.3 | 42.7 / 33.5 | 55.4 / 40.1 | 69.5 / 49.6 | 58.0 / 48.3 | 64.5 / 49.4 |
| XLM | 74.2 / 62.1 | 61.4 / 44.7 | 66.0 / 49.7 | 57.5 / 39.1 | 68.2 / 49.8 | 56.6 / 40.3 | 65.3 / 48.2 | 35.4 / 24.5 | 57.9 / 41.2 | 65.8 / 47.6 | 49.7 / 39.7 | 59.8 / 44.3 |
| MMTE | 80.1 / 68.1 | 63.2 / 46.2 | 68.8 / 50.3 | 61.3 / 35.9 | 72.4 / 52.5 | 61.3 / 47.2 | 68.4 / 45.2 | 48.4 / 35.9 | 58.1 / 40.9 | 70.9 / 50.1 | 55.8 / 36.4 | 64.4 / 46.2 |
| XLM-R | 86.5 / 75.7 | 68.6 / 49.0 | 80.4 / 63.4 | 79.8 / 61.7 | 82.0 / 63.9 | 76.7 / 59.7 | 80.1 / 64.3 | 74.2 / 62.8 | 75.9 / 59.3 | 79.1 / 59.0 | 59.3 / 50.0 | 76.6 / 60.8 |
| Filter | 86.4 / 74.6 | 79.5 / 60.7 | 83.2 / 67.0 | 83.0 / 64.6 | 85.0 / 67.9 | 83.1 / 66.6 | 82.8 / 67.4 | 79.6 / 73.2 | 80.4 / 64.4 | 83.8 / 64.7 | 79.9 / 77.0 | 82.4 / 68.0 |
| xTune | 88.8 / 78.1 | 79.7 / 63.9 | 83.7 / 68.2 | 83.0 / 65.7 | 84.7 / 68.3 | 80.7 / 64.9 | 82.2 / 66.6 | 81.9 / 76.1 | 79.3 / 65.0 | 82.7 / 64.5 | 81.3 / 78.0 | 82.5 / 69.0 |
| X-Mixup | 86.7 / 75.4 | 81.3 / 63.5 | 83.5 / 66.8 | 84.3 / 67.6 | 85.2 / 68.2 | 83.9 / 68.5 | 83.0 / 67.7 | 82.6 / 76.9 | 80.9 / 65.3 | 84.8 / 66.8 | 72.4 / 75.6 | 82.6 / 69.3 |

Table 15: XQuAD results (F1 / EM) for each language.

| Model | en | ar | de | es | hi | vi | zh | Avg. |
|---|---|---|---|---|---|---|---|---|
| mBERT | 80.2 / 67.0 | 52.3 / 34.6 | 59.0 / 43.8 | 67.4 / 49.2 | 50.2 / 35.3 | 61.2 / 40.7 | 59.6 / 38.6 | 61.4 / 44.2 |
| XLM | 68.6 / 55.2 | 42.5 / 25.2 | 50.8 / 37.2 | 54.7 / 37.9 | 34.4 / 21.1 | 48.3 / 30.2 | 40.5 / 21.9 | 48.5 / 32.6 |
| MMTE | 78.5 / – | 56.1 / – | 58.4 / – | 64.9 / – | 46.2 / – | 59.4 / – | 58.3 / – | 60.3 / 41.4 |
| XLM-R | 83.5 / 70.6 | 66.6 / 47.1 | 70.1 / 54.9 | 74.1 / 56.6 | 70.6 / 53.1 | 74.0 / 52.9 | 62.1 / 37.0 | 71.6 / 53.2 |
| Filter | 84.0 / 70.8 | 72.1 / 51.1 | 74.8 /60.0 | 78.1 / 60.1 | 76.0 / 57.6 | 78.1 /57.5 | 70.5 / 47.0 | 76.2 / 57.7 |
| xTune | 85.3 / 72.9 | 69.7 / 50.1 | 72.3 / 57.3 | 76.3 / 58.8 | 74.0 / 56.0 | 76.5 / 55.9 | 70.8 / 48.3 | 75.0 / 57.1 |
| X-Mixup | 83.1 / 70.0 | 71.9 / 51.1 | 74.5 / 59.4 | 77.7 / 60.0 | 76.3 / 57.7 | 78.0 / 57.5 | 73.7 / 51.1 | 76.5 / 58.1 |

Table 16: MLQA results (F1 / EM) for each language.

| Model | en | ar | bn | fi | id | ko | ru | sw | te | Avg. |
|---|---|---|---|---|---|---|---|---|---|---|
| mBERT | 75.3 / 63.6 | 62.2 / 42.8 | 49.3 / 32.7 | 59.7 / 45.3 | 64.8 / 45.8 | 58.8 / 50.0 | 60.0 / 38.8 | 57.5 / 37.9 | 49.6 / 38.4 | 59.7 / 43.9 |
| XLM | 66.9 / 53.9 | 59.4 / 41.2 | 27.2 / 15.0 | 58.2 / 41.4 | 62.5 / 45.8 | 14.2 / 5.1 | 49.2 / 30.7 | 39.4 / 21.6 | 15.5 / 6.9 | 43.6 / 29.1 |
| MMTE | 62.9 / 49.8 | 63.1 / 39.2 | 55.8 / 41.9 | 53.9 / 42.1 | 60.9 / 47.6 | 49.9 / 42.6 | 58.9 / 37.9 | 63.1 / 47.2 | 54.2 / 45.8 | 58.1 / 43.8 |
| XLM-R | 71.5 / 56.8 | 67.6 / 40.4 | 64.0 / 47.8 | 70.5 / 53.2 | 77.4 / 61.9 | 31.9 / 10.9 | 67.0 / 42.1 | 66.1 / 48.1 | 70.1 / 43.6 | 65.1 / 45.0 |
| Filter | 72.4 / 59.1 | 72.8 / 50.8 | 70.5 / 56.6 | 73.3 / 57.2 | 76.8 / 59.8 | 33.1 / 12.3 | 68.9 / 46.6 | 77.4 / 65.7 | 69.9 / 50.4 | 68.3 / 50.9 |
| xTune | 73.8 / 61.6 | 77.8 / 60.2 | 73.5 / 61.1 | 77.0 / 62.2 | 80.8 / 68.1 | 66.9 / 56.5 | 72.1 / 51.9 | 77.9 / 65.3 | 77.6 / 60.7 | 75.3 / 60.8 |
| X-Mixup | 73.9 / 61.4 | 73.8 / 54.2 | 67.4 / 49.6 | 75.4 / 60.6 | 78.8 / 65.0 | 32.9 / 12.0 | 69.1 / 52.2 | 78.0 / 66.9 | 72.0 / 53.5 | 69.0 / 52.8 |

Table 17: TyDiQA-GolP results (F1 / EM) for each language.

