# OpenReview forum: "Enhancing Cross-lingual Transfer by Manifold Mixup"
_ICLR.cc/2022/Conference — ICLR 2022 Poster_

### Official Review · Reviewer_pZJi · 2021-10-29

**Correctness:** 4
**Technical Novelty And Significance:** 3
**Empirical Novelty And Significance:** 3
**Recommendation:** 8
**Confidence:** 5

**Main Review:**

Strengths

1. The paper is well written and the motivation is clear. The paper first conducts some priori experiments to strength the motivation, and transforms the performance gap to the representation discrepancy through the visiualization analysis.
2. The manifold mixup seems to be valuable for bridging the cross-lingual representation discrepancy across languages.
3. The improvements on XTREME over XLM-R is pretty strong.

Cons:

1. What does $\tilde{\mathcal{D}}_{S}^{Train}$ means? I didn't find its definition in the "Notations" part.
2. I would like to see ablations for the mixup ratio $\lambda$ where you use randomly sampled value for $\lambda$ or compute a gate using $h_{T|S}$ and $h_{T}$, rather depend on $A$.
3. Why you need the consistency loss in the training objective? What does $P_S$ and $P_T$ exactly mean? How to deal with the KL collapse issue?
4. What does $\tilde{D}_S$ (appeared in the Scheduled Sampling and Inference parts) means?
5. The scheduled sampling strategy has a great impact on training efficiency. Can you provide the training time of the X-Mixup and translate-train respectively?

**Summary Of The Paper:**

The work applies and adjusts manifold mixup in the subject area of zero-shot cross-lingual transfer. The work first identifies the challenges with the current landscape of cross-lingual transfer methods, in which the performance of the target languages still lags for far behind the source language. The paper make the gap associated with cross-lingual representation discrepancy. To take care this, the authors propose a X-Mixup method that interpolates the representations between the source and target languages, and give this compromised representation for target language prediction. Compared to strong baselines, the work obtains significant gains on the XTREME benchmark.

**Summary Of The Review:**

I score this paper a 6. I think the manifold mixup is very interesting for this use-case and made my reading and evaluation of this paper more interesting. I look forward to the responses from the authors.

---

> ### Author Response · Authors · 2021-11-16
> **Response to Reviewer pZJi**
>
>
> Thanks for your insightful feedback and constructive suggestions. We have fixed uncleared notations, statements, and add analyses according to your suggestions in the new version.
>
> Here are the replies to each of your question/suggestion:
>
> [**Notations of $\tilde D_{S}$ and $\tilde D_{S} ^ {Train}$**]
>
> Sorry for the confusing notations. We have made them clear in the new version.
>
> $\tilde D_{S}$ denotes the translation data in the source language.
> During training, $\tilde D_{S}$, i.e. $\tilde D_{S} ^ {Train}$, is the translation data in the source language, which are acquired by forward translation (from $S$ to $T$) then backward translation (from $T$ to $S$).
> During inference,  $\tilde D_{S}$, i.e. $\tilde D_{S} ^ {Test}$, is the translate-test data, which are acquired by translating test data in the target language to the source language.
>
> [**Ablation of mixup ratio**]
>
> It is a very good suggestion to implement ablation studies on the MT quality estimation, which uses $\lambda=\lambda_0$, a constant mixup ratio.  We list ablation results in the following table, which show the effectiveness of introducing the translation quality modeling in the mixup process on three tasks. We have added these results in Table 6 of the new version.
>
> | Method | XNLI | POS | MLQA |
> |----| ---- | ---- | ---- |
> | X-Mixup | 85.3 | 78.4 | 76.5/58.1 |
> | $\lambda=\lambda_0$ | 84.1 |  77.8 | 75.8/57.5 |
>
> [**Consistency loss**]
>
> The representation consistency loss, i.e. MSE loss, aims at aligning representations across different languages.
> The prediction consistency loss of the classification task, i.e. KL loss, aims at ensuring the consistency of prediction across different languages to better utilize the supervision of downstream tasks, where $\mathbf{p}_S$ and $\mathbf{p}_T$ are predicted probability distributions on downstream tasks of the source sequence and target sequence respectively.
> Consistency loss is joint training with task loss, where the gold label is the one-hot label. Besides, the total objective is not the ELBO objective. As a result, the KL (posterior) collapse issue will not appear in the mixup training process.
>
> [**Training time**]
>
> For each downstream task, X-Mixup and Translate-train method train the model for the same step. The Scheduled Sampling strategy in X-Mixup samples the source sequence dynamically (either a real text or translation), which does not increase the overall training step. For each training step, the computation amount of X-Mixup doubles based on Translate-train. However, the source sequence and target sequence can be encoded simultaneously, and the actual training speed of X-Mixup is about 1.08 times the speed of Translate-train.

---

> > ### Author Response · Authors · 2021-11-22
> > **Continue response to Reviewer pZJi**
> >
> > We really hope our response has addressed your concern. If you have any other questions or suggestions, we are willing to discuss them further.

---

> > > ### Comment · Reviewer_pZJi · 2021-11-24
> > > **Reply to response**
> > >
> > > I would like to thank to authors for addressing my feedback, and my questions are properly resolved. I will increase my score from 6 to 8.

---

### Official Review · Reviewer_DF3R · 2021-11-02

**Correctness:** 4
**Technical Novelty And Significance:** 4
**Empirical Novelty And Significance:** 3
**Recommendation:** 6
**Confidence:** 4

**Main Review:**

Overall, I like this idea that considers the source languages and the target languages at the same time to encourage the model to learn more similar representations for different languages. Here are some suggestions and questions:
- Figure 4 only contains a few languages and they are mostly high-resource languages. It would be more convincing if the authors can show results of more languages, especially low-resource languages.
- What if there are multiple source languages? Do we randomly sample one of them during training and testing?
- It seems like we need to know what target languages we will care about in advance. What if there are some new target languages? Some analysis on this can be quite interesting. For example, only consider part of target languages during training and see if those languages help the model learn good representations for unseen languages as well.
- Why the consistency loss consists of two terms: MSE and KL. Any ablations on this?
- Does more CKA score improvement lead to more task performance improvement? Any relations between them?


**Summary Of The Paper:**

This paper proposes X-Mixup, a model that considers the source languages and target languages together for cross-lingual transfer. The designed model takes a pair of sentences (or the translated sentences) in a source language and a target language as the input and computes the cross-attention between them. The author also adds the consistency loss to encourage similar predictions for the source and the target. They also propose scheduled sampling to handle the different distributions during training and testing. The experimental results and the ablation studies show promising performances for cross-lingual transfer.

**Summary Of The Review:**

The proposed idea is interesting. They show good and promising experimental results and ablation studies. Some questions need to be clarified and some more analysis is suggested.

---

> ### Author Response · Authors · 2021-11-16
> **Response to Reviewer DF3R**
>
> Thanks for your insightful feedback and constructive suggestions. We have added experiments and fixed unclear statements according to your suggestion in the new version.
>
> Here are the replies to each of your question/suggestion:
>
> [**Figure 4 should cover more languages, especially low-resource languages**]
>
> It is a very good suggestion to show more languages in Figure 4. We have updated Figure 4 (Figure 3 in the new version), including both high-resource languages like es, fr and low-resource languages like sw, ur, which shows the  cross-lingual transfer performance is getting worse along with the CKA score decreasing, regardless of the language resource.
>
> [**Multiple source languages setting**]
>
> Our experiments are based on the XTREME benchmark, where English is the only source language. For the multi-source setting, the simplest way of implementing X-Mixup is to select one of the source languages randomly as the source language. However, different target languages tend to show different relevance to each source language, where the cross-lingual representation discrepancy could be complex (the number of source languages multiplies the number of target languages). X-Mixup can be expanded to the Mix-of-Expert style, where each target sequence can be mixed with every source sequence based on the normalized Mixup Ratio. We are glad to expand X-Mixup to the multi-source setting in the future.
>
> [**New target languages**]
>
> It is a very good suggestion to consider new target languages which are unseen during training. We randomly select 7 languages (ar, de, es, hi, sw, tr, vi) out of 15 languages in XNLI as new target languages. Results in Table 7 of the new version verify the effectiveness of X-Mixup on both seen and unseen languages.
>
> Interestingly, comparing results in Table 8 and Table 11, we find finetuning on 8 out of 15 languages performs on par with finetuning on 15 languages in XNLI, which implies the translate-train data of unseen languages does not make a lot of contributions during finetuning. Data/language selection is also an interesting and promising direction in cross-lingual transfer. We are glad to focus on this direction in the future.
>
> [**Ablation results of consistency loss**]
>
> The representation consistency loss, i.e. MSE loss, aims at aligning representations across different languages.
> The prediction consistency loss of the classification task, i.e. KL loss, aims at ensuring the consistency of prediction across different languages to better utilize the supervision of downstream tasks. This loss only exists in the classification task, as the translation process does not change the label of this task, while in other tasks, it does.
>
> We implement the ablation study on consistency loss in the following table, which shows the KL consistency loss contributes more than the MSE consistency loss on the classification task XNLI. We have added these ablation results in the new version.
>
> |Method | XNLI |  POS | MLQA |
> |----| ---- | ---- | ---- |
> | X-Mixup | 85.3 | 78.4 | 76.5/58.1 |
> | $\quad$ w/o MSE loss | 84.6 | 78.0 | 76.5/58.0 |
> | $\quad$ w/o KL loss | 84.3 |  - | - |
> | $\quad$ w/o both | 84.2 |  - | - |
>
> [**Relation between the CKA score improvement and performance improvement**]
>
> Table 8 (in the new version) shows the CKA score and performance of each language. Results show X-Mixup improves the CKA score evenly across different target languages. But the performance improvement is not evenly distributed. There is no obvious correlation between the CKA score improvement and performance improvement.

---

> > ### Comment · Reviewer_DF3R · 2021-11-21
> > **Reply to response**
> >
> > Thanks for the response. My questions are properly resolved. On suggestion is that listing all languages for XNLI in Figure 4 (Figure 3 in the new version) so we can observe the trend more easily.

---

> > > ### Author Response · Authors · 2021-11-22
> > > **Continue response to Reviewer DF3R**
> > >
> > > Thanks for your valuable suggestion. We have updated Figure 3 in the new version according to your suggestion. Both the trend in Figure 3 and the correlation score in Table 2 confirm the cross-lingual transfer performance is strongly associated with the cross-lingual representation discrepancy.
> > >
> > > If you have any other questions or suggestions, we are willing to discuss them further.

---

### Official Review · Reviewer_9Shh · 2021-11-03

**Correctness:** 3
**Technical Novelty And Significance:** 3
**Empirical Novelty And Significance:** 2
**Recommendation:** 5
**Confidence:** 3

**Main Review:**

This paper proposes a technique called *cross-lingual manifold mixup* or *X-mixup*, which has been inspired by *mixup* (Zhang et al., 2018) and *Filter* (Fang et al., 2020). The general idea is to combine the hidden representations corresponding to the source-language input with the hidden representations corresponding to the target-language input (in a smarter way than, e.g., simply concatenating them). The authors combine this with 1) a specific mixup ratio based on translation quality and 2) scheduled sampling. They perform multiple experiments using two pretrained models (mBERT and XLM-R) and show that their proposed approach improves over multiple strong baselines.

Strengths:
- The approach works: the authors show improved results over multiple strong baselines.
- The paper is well written and the authors perform a nice analysis/investigation of smaller research questions in Section 6.

Weaknesses:
- I am unsure if framing *X-mixup* as a cross-lingual transfer version of *mixup* is the best framing. I only read the *mixup* paper superficially, but it seems to me that that paper's goal is to generate additional training examples by averaging (independent) pairs for both *x* and *y* to make the model more robust. In contrast, what the authors do here is more closely related to simply concatenating source and target sentence in the input.
- The experiment section isn't very clean. For example, it is unclear why different methods are used on top of mBERT and XLM-R, respectively (see Table 3).

**Summary Of The Paper:**

This paper proposes a technique which the authors call *cross-lingual manifold mixup* or *X-mixup*. The approach has been inspired by *mixup* (Zhang et al., 2018) and *Filter* (Fang et al., 2020). The general idea is to combine the hidden representations corresponding to the source-language input with the hidden representations corresponding to the target-language input (in a smarter way than, e.g., simply concatenating them). The authors perform multiple experiments using two pretrained models (mBERT and XLM-R) and show that their proposed approach improves over multiple strong baselines.

**Summary Of The Review:**

The paper presents *X-mixup*, a method for cross-lingual transfer based on translate-train, which improves performance over multiple strong baselines. However, I have doubts regarding the framing of the paper as well as regarding the conducted experiments. Thus, I think the paper should be revised before being published.

---

> ### Author Response · Authors · 2021-11-15
> **Response to Reviewer 9Shh**
>
> Thanks for your valuable feedback. We have revised our draft according to suggestions and questions from reviewers. Please take a look. Could you please point out other unclear parts in the method or experiment section? We appreciate your help in advance and we will make them clear as soon as possible.
>
> Here are the replies to each of your question/comment:
>
> [**X-Mixup framework**]
>
> (1) X-Mixup borrows the interpolation idea from Mixup, but it is more suitable for the cross-lingual transfer setting.
>
> Mixup (Zhang et al., 2018; Verma et al., 2019) focuses on improving the robustness of model by generating additional training examples through the interpolation operation, which is simple but really works in downstream tasks. However, X-Mixup focuses on reducing the cross-lingual representation discrepancy by giving compromised representations for target languages. In the cross-lingual transfer setting, we find the cross-lingual transfer performance is strongly associated with the cross-lingual representation discrepancy. Based on this, X-Mixup gives compromised representations for target languages by mixing the representation of the source and target languages, which reduces the cross-lingual representation discrepancy distinctly.
>
> (2) In the cross-lingual transfer setting, X-Mixup faces many new challenges.
>
> X-Mixup is designed upon the translate-train approach, faced with the exposure bias problem and data noise problem. During training, the source sequence is a real sentence and the target sequence is a translated one, while situations are opposite during inference. Besides, the translated text often introduces some noises due to imperfect machine translation systems. X-Mixup further imposes the Scheduled Sampling and Mixup Ratio to handle the distribution shift problem and data noise problem, respectively.
>
> (3) X-Mixup is not simply concatenating the source and target sentence but gives calibrated representations for target sentences.
>
> Filter (Fang et al. 2020) implements the input / hidden states concatenation of the source and target sequences. However, X-Mixup implements the hidden states mixup based on the learned Mixup Ratio, which gives a calibrated representation for target languages, not just the concatenation of two representations. Besides, X-Mixup considers the noise of translation data and limits the noise propagation by the learnable Mixup Ratio. Results in Table 3 show X-Mixup outperforms Filter by 1.5%, which indicates the superiority of X-Mixup over concatenating.
>
> [**Backbone model**]
>
> XLM-R and mBERT are two commonly used backbone models in cross-lingual understanding tasks. Filter (Fang et al. 2020) is based on the XLM-R model and Joint-Align (Zhao et al. 2020) is based on the mBERT model. To verify the generality of X-Mixup, we implement experiments on both XLM-R and mBERT. In Table 3, X-Mixup shows its superiority across different backbones and tasks, which indicates its generality.

---

> > ### Author Response · Authors · 2021-11-22
> > **Continue response to Reviewer 9Shh**
> >
> > We really hope our response has addressed your concern. If you have any other questions or suggestions, we are willing to discuss them further.

---

### Official Review · Reviewer_VD9r · 2021-11-05

**Correctness:** 4
**Technical Novelty And Significance:** 3
**Empirical Novelty And Significance:** 3
**Recommendation:** 8
**Confidence:** 4

**Main Review:**


Strengths:
- x-mixup is a clear idea and seems straightforward to include on top of existing models like mBERT or XLM-R.
- The improvements over strong baselines on multiple XTREME tasks are convincing.

Weaknesses:
- One big downside I see is that the authors use mixup at different layers for different tasks. What happens if a single setting is picked and used for all tasks? How does the comparison look?
- The regularization term to account for MT quality should be further tested by controlling the quality of MT and seeing if it's robust to it. Otherwise it's not clear how valuable it is.
- Another downside is that this method relies on translation data (both translate-train and translate-test) during training. I would like to see an ablation where only translate-train data is used or only translate-test data is used. This would be fair comparison to the XTREME submissions which only use translate-train and not translate-test.


Detailed comments:
- The related work section is a bit weak.
- Some missing citations:
     - there's a whole body of work on using translation data for cross-lingual learning: please cite
                 VECO: Variable and Flexible Cross-lingual Pre-training for Language Understanding and Generation
                 Evaluating the Cross-Lingual Effectiveness of Massively Multilingual Neural Machine Translation
                 Explicit Alignment Objectives for Multilingual Bidirectional Encoders
                 nmT5 -- Is parallel data still relevant for pre-training massively multilingual language models?
- Footnote #9 is a very important detail. It should be moved to the main paper.
- Some minor grammar mistakes but overall clear paper.
- Please have a discussion on how this be applied to the retrieval tasks in XTREME. Can this approach be moved to the pre-training set up by leveraging some parallel data?
- Please also discuss the impact on training time by adding x-mixup compared to translate-train.



**Summary Of The Paper:**

The paper proposes cross-lingual mixup, a technique which performs manifold mixup of source and target sequences. This technique also includes mixup ratio which factors in MT quality and scheduled sampling which deals with exposure bias. Experiments on 3 task types from the XTREME benchmark show that x-mixup leads to significant gains over translate-train baselines and other strong submissions to the leaderboard. The authors also present some analyses with a few ablations.

**Summary Of The Review:**

The authors propose x-mixup a technique which leverages parallel data and forces cross-lingual representations to be aligned. The proposed method leverages translation data (translate-train and translate-test) during training. This leads to improvements in several XTREME downstream tasks over strong baselines. The authors perform some ablations which showcase the importance of the different components in the proposed method.

---

> ### Author Response · Authors · 2021-11-16
> **Response to Reviewer VD9r**
>
> Thanks for your insightful feedback and constructive suggestions. We have clarified unclear statements and added experiments and references according to your suggestions in the new version.
>
> Here are the replies to each of your question/suggestion:
>
> [**Mixup layer**]
>
> It is a very good suggestion to fix the mixup layer on different tasks. Figure 5 shows performances of mixup in different layers on three tasks. Although different tasks have their own preferences, no matter which layer we mix (fix the x-axis in Figure 5), the cross-lingual transfer performance can be improved, except for mixing within a higher layer on classification task. The drop in classification task is mainly because the source and target sequences share the same task label. Performing mixup in a higher layer can make the model rely on the source sequence and ignore the target sequence. We have added relevant analyses in the new version.
>
> [**The necessity of introducing MT quality estimation in X-Mixup**]
>
> It is also a very good suggestion to implement ablation studies on the MT quality estimation, which uses $\lambda=\lambda_0$, a constant mixup ratio.  We list ablation results in the following table, which show the effectiveness of introducing the translation quality modeling in the mixup process on three tasks. We have added these results in Table 6 of the new version.
>
> | Method | XNLI | POS | MLQA |
> |----| ---- | ---- | ---- |
> | X-Mixup | 85.3 | 78.4 | 76.5/58.1 |
> | $\lambda=\lambda_0$ | 84.1 |  77.8 | 75.8/57.5 |
>
> [**Ablation results of without translate-test data**]
>
> In Table 6, "w/o mixup inference" denotes the monolingual inference without translate-test data. Results show the mixup training can achieve 2.6% improvements on average compared with Translate-train, even without mixup inference (translate-test).
>
> [**Missing citations**]
>
> We really appreciate that you pointed out these missing references. These pre-training methods propose new pre-training objectives to make better use of parallel data, which achieve significant improvements on downstream tasks. We have added these missing references and enhanced the related work section in the new version.
>
> [**Apply X-Mixup to retrieval tasks and pre-training process**]
>
> (1) X-Mixup on retrieval tasks
>
> Based on pre-trained multilingual representations (without downstream task finetuning), for cross-lingual sentence retrieval tasks (Tatoeba, BUCC), the target language sequence can be mixed with its translation based on the constant Mixup Ratio ($\lambda=\lambda_0$). Both Figure 1, 4 and CKA scores in Table 8 show X-Mixup reduces the cross-lingual representation discrepancy obviously, which implies the potential of X-Mixup on cross-lingual sentence retrieval tasks.
>
> Filter (Fang et al., 2020) implements the cross-lingual retrieval task based on the model finetuned on XNLI. With downstream task finetuning, in cross-lingual sentence retrieval tasks, the target language sequence can be mixed with its translation based on the learned Mixup Ratio, which takes the translation quality into account.
>
> (2) X-Mixup in pre-training process
>
> X-Mixup provides an effective way to use parallel or pseudo-parallel data to reduce the cross-lingual representation discrepancy. We can also implement the X-Mixup method during pre-training by mixing parallel or pseudo-parallel sequences. It is a very promising direction and we are glad to apply X-Mixup to multilingual pre-training in the future.
>
> [**Training time**]
>
> For each downstream task, X-Mixup and Translate-train method train the model for the same step. The Scheduled Sampling strategy in X-Mixup samples the source sequence dynamically (either a real text or translation), which does not increase the overall training step. For each training step, the computation amount of X-Mixup doubles based on Translate-train. However, the source sequence and target sequence can be encoded simultaneously, and the actual training speed of X-Mixup is about 1.08 times the speed of Translate-train.

---

> > ### Author Response · Authors · 2021-11-22
> > **Continue response to Reviewer VD9r**
> >
> > We really hope our response has addressed your concern. If you have any other questions or suggestions, we are willing to discuss them further.

---

> > > ### Comment · Reviewer_VD9r · 2021-11-29
> > > **Response to author rebuttal**
> > >
> > > Thanks for addressing some of the comments. I think the new additions definitely improve the paper.

---

### Author Response · Authors · 2021-11-15
**Paper Revision**

We sincerely thank all reviewers for their insightful comments. We have uploaded a new version of our draft, clarifying some statements and adding more analyses:
- Add missing citations in the Related Work section (Section 2, Page 2).
- Add more ablation experiments on X-Mixup, including the Mixup Ratio ablation experiment (Table 6, Page 9) and the consistency loss ablation experiment (Table 9, Page 17).
- Clarify statements of Notations (Section 4.1, Page 4) and Mixup layer (Section 6 Q4, Page 9).
- Add more languages in Figure 3, including both high-resource and low-resource languages (Page 16).
- Add new target languages experiments in Table 7 (Page 16).
- Add detailed improvements on each language in Table 8 (Page 16).

 Please take a look!

---

### Decision · Program_Chairs · 2022-01-20

**Decision:**

Accept (Poster)

**Comment:**

This paper proposes X-Mixup, a model that considers the source languages and target languages together for cross-lingual transfer. The designed model takes a pair of sentences (or the translated sentences) in a source language and a target language as the input and computes the cross-attention between them.

The empirical results are convincing. Reviewers think this paper is well-written and the idea is interesting.